# Dynamic Alterations in Acetylation and Modulation of Histone Deacetylase Expression Evident in the Dentine–Pulp Complex during Dentinogenesis

**DOI:** 10.3390/ijms25126569

**Published:** 2024-06-14

**Authors:** Yukako Yamauchi, Emi Shimizu, Henry F. Duncan

**Affiliations:** 1Division of Restorative Dentistry & Periodontology, Dublin Dental University Hospital, Trinity College Dublin, Lincoln Place, D02 F859 Dublin, Ireland; yamauchi.yukako.dent@osaka-u.ac.jp; 2Department of Oral Biology, Rutgers School of Dental Medicine, Newark, NJ 07103, USA; shimize1@sdm.rutgers.edu

**Keywords:** pulp biology, histone deacetylases, tooth development, epigenetics, dentinogenesis, odontoblasts

## Abstract

Epigenetic modulation, including histone modification, alters gene expression and controls cell fate. Histone deacetylases (HDACs) are identified as important regulators of dental pulp cell (DPC) mineralisation processes. Currently, there is a paucity of information regarding the nature of histone modification and HDAC expression in the dentine–pulp complex during dentinogenesis. The aim of this study was to investigate post-translational histone modulation and HDAC expression during DPC mineralisation and the expression of Class I/II HDACs during tooth development and in adult teeth. HDAC expression (isoforms −1 to −6) was analysed in mineralising primary rat DPCs using qRT-PCR and Western blot with mass spectrometry being used to analyse post-translational histone modifications. Maxillary molar teeth from postnatal and adult rats were analysed using immunohistochemical (IHC) staining for HDACs (1–6). HDAC-1, -2, and -4 protein expression increased until days 7 and 11, but decreased at days 14 and 21, while other HDAC expression increased continuously for 21 days. The Class II mineralisation-associated HDAC-4 was strongly expressed in postnatal sample odontoblasts and DPCs, but weakly in adult teeth, while other Class II HDACs (-5, -6) were relatively strongly expressed in postnatal DPCs and adult odontoblasts. Among Class I HDACs, HDAC-1 showed high expression in postnatal teeth, notably in ameloblasts and odontoblasts. HDAC-2 and -3 had extremely low expression in the rat dentine–pulp complex. Significant increases in acetylation were noted during DPC mineralisation processes, while trimethylation H3K9 and H3K27 marks decreased, and the HDAC-inhibitor suberoylanilide hydroxamic acid (SAHA) enhanced H3K27me3. These results highlight a dynamic alteration in histone acetylation during mineralisation and indicate the relevance of Class II HDAC expression in tooth development and regenerative processes.

## 1. Introduction

The recent promotion of the development of biologically based vital pulp treatments (VPTs) for the treatment of the injured pulp has stimulated interest in developing targeted restorative solutions that may promote hard tissue formation and modulate inflammation in the dental pulp [1]. Identifying novel key molecules in pulpitis and pulpal repair enhances understanding and assists the development of next-generation materials that focus on dental repair or act as immunotherapeutic agents [2]. Previous research has highlighted the potential utility of antioxidants [3], antihypertensives [4] and epigenetic modifiers [5] to promote reparative processes in the dental pulp cell (DPC) populations. The principal epigenetic modifications are DNA methylation and DNA-associated histone acetylation. Although the inhibition of both processes has stimulated therapeutic attention [6,7], histone modifications [8,9], and specifically histone acetylation, have attracted the most attention.

Histone acetylation is regulated with two opposing cellular enzymes, histone acetyltransferases (HATs) and histone deacetylases (HDACs). While HAT activity creates a relaxed transcriptionally active chromatin structure, HDAC activity has the opposite effect [10]. Eighteen human HDAC enzymes have been placed in four separate classes, with Classes I, II, and IV containing the zinc-dependent enzymes [11] and Class III the Sirtuins [12]. Class I HDACs are ubiquitously expressed in the cell nucleus, while Class II HDACs vary in expression depending on the needs of the tissue and are placed in the cytoplasm and nucleus [13]. The role and expression of Class II HDAC expression in mineralising tissues have been previously demonstrated in bone [14] with specific isoforms, −6 [15], −5, and −4 [16], highlighted as being important cellular mediators that regulate osteoblast differentiation and may present as specific targets for selective inhibition that can be turned on and off using inhibitors. Within VPT procedures such as pulp capping or pulpotomy, the potential use of pharmacological inhibitors that can regulate odontoblast differentiation and mineralisation processes in a controlled manner is of fundamental importance in developing smarter restorative solutions [1,2], as uncontrolled repair or dystrophic mineralisation responses may hinder repair and complicate subsequent clinical management.

HDAC expression alterations occur during osteogenesis [15,17], primary dentinogenesis [18], and cementogenesis [19] in a tissue-specific manner. Class I and II HDAC expression analysed in human tooth periodontal ligament cell (PDLC) cultures demonstrated that all of the five HDACs studied (HDAC-1 to -4 and -6) were highly expressed, although HDAC-3 was downregulated during osteogenic differentiation [19]. Furthermore, a dental pulp study analysing extracted adult human molar teeth demonstrated that HDAC-2 and -9 were expressed in DPC and exhibited a relatively strong expression in odontoblasts, while HDAC-1, -3, and -4 were relatively weakly expressed within the pulp tissue [18].

Clearly, epigenetic modulations, DNA-methylation, and histone modifications are important regulators of dental stem cell fate [20] as well as dental pulp mineralisation processes and therefore present an attractive therapeutic drug target [7,19]. Current HDAC inhibitor (HDACi)studies tend to use pan-HDACi, such as suberoylanilide hydroxamic acid (SAHA), valproic acid, and trichostatin A [5,7,21], and although these studies have demonstrated positive results, there is concern that their broad effect may have off-site targets and nonspecific actions [22]. As a result, therapeutic strategies targeting one or two HDAC isoforms using isoform-specific inhibitors have been promoted [23]. However, in order for these inhibitors to be effective in strategies to manage the dental pulp, the expression of Class I and II HDACs will need to be established during mineralisation in the dentine–pulp complex.

As the dentine pulp expression of Class I and II HDACs has not been established in developing teeth compared with adult teeth, the aim of the current study was to investigate in vitro HDAC expression during DPC mineralisation and the in vivo expression of Class I/II HDACs during rat tooth development (primary dentinogenesis) as well as in adult rat teeth (secondary dentinogenesis). A further aim was to study the nature of a wide range of post-translational histone modifications during mineralising in a DPC population. We hypothesised that HDAC isoforms 1–6 would alter in expression during a 21-day mineralisation cycle in vitro and that the expression of Class II HDAC isoforms would be increased in expression during primary dentinogenesis. Furthermore, we hypothesised that a range of characterised and uncharacterised epigenetic marks would be altered during DPC mineralisation.

## 2. Results

In a series of linked experiments, the gene and protein expression of HDAC1–6 were investigated in rat DPCs in vitro in mineralising and non-mineralising cultures. Thereafter, the in vivo expression of the same HDAC isoforms was analysed via immunohistochemistry (IHC) in developing and mature rodent tooth models. Finally, the specific post-translational modifications occurring during mineralisation were investigated in DPCs in vitro (Figure 1).

### 2.1. Class II HDAC-4 Showed Increased Expression during DPC Mineralisation In Vitro with HDAC-6 Showing Decreased Gene Expression

A time-course analysis using qRT-PCR showed that *HDAC-1*, *-2*, *-3*, and *-5* did not significantly change in expression over the 14 days, while *HDAC-4* significantly increased in expression at 14 days (*p* < 0.05) and *HDAC-6* significantly decreased in expression at days 8 and 14 (both *p* < 0.05) (Figure 2). Class I HDAC (-1, -2) and Class II HDAC (-4) protein expression increased until peaking at day 7, before decreasing towards day 21 (Figure 3A–C). Semi-quantitative Western blot analysis revealed that HDAC-2 increased by an average of 5-fold at 7 days, with the fold increase for HDAC-1 and -4 not exceeding 2.5-fold. The Class II HDAC-6 showed a general increase in protein expression in DPC expression after 1 day until 21 days (Figure 3D). HDAC-3 and -5 both showed very low levels of DPC expression (Figure 3E,F). Mineralisation was confirmed via alizarin red staining as initiated on day 7 and increasing in intensity until day 21 (Figure 3G).

### 2.2. HDAC-1 Isoforms Exhibit Altered Expression in Postnatal and Adult Molar Teeth

Class I HDAC expression in rat maxillary molars at postnatal day 5 (Figure 4A,C,E) was histologically compared with adult maxillary molar teeth at 8 weeks (Figure 4B,D,F). HDAC-1 was highly expressed in postnatal dental pulp, with notable nuclear expression in odontoblasts and the subodontoblast zone (Figure 4A) compared with the negative control. HDAC-1 nuclear expression was still evident but reduced in adult teeth (Figure 4B), which is suggestive of a potential role for HDAC-1 in dentinogenesis. The in vivo dentine pulp expression of HDAC-2 and -3 were low in both developing and adult teeth (Figure 4E,F), supporting protein expression results in vitro (Figure 3).

### 2.3. Class II HDAC-4 and -6 Showed High Expression during Tooth Development with HDAC-4 Reducing in Mature Teeth

Class II HDAC isoforms have previously been shown to play an important role in osteogenesis [15,16]. The mineralisation-associated HDAC-4 was strongly expressed in postnatal odontoblasts and DPCs, but weakly in adult teeth, supporting both the protein and mRNA results reported earlier (Figure 5A). HDAC-4 expression was generally strong in developing teeth compared with the rest of the oral tissue visible on the slide (Figure 5Ai). Other Class II HDACs (-5, -6) were relatively strongly expressed in postnatal DPCs and adult odontoblasts with a suggestion that HDAC-6 was increased in expression in adult compared with postnatal samples (Figure 5D,F).

### 2.4. Post-Translational Histone Modification Alters Significantly during Mineralisation

To assess the broader landscape of the epigenetic regulation of DPC mineralisation processes via histone modification, a mass spectrometry-based screen for over 80 histone modifications in non-induced, induced (mineralisation media), and induced + SAHA cultures at 5 days was performed. The five-day timepoint was chosen to visualise relatively early changes during mineralisation. This screen allowed the visualisation of not only the acetylation but also the methylation and ubiquitination of peptide residues during mineralisation processes in DPCs. Three DPC culture groups were analysed, including a non-mineralising control group (non-induced), a mineralising group (induced), and a mineralising group in the presence of the HDACi, SAHA. When the modification levels fell below 0.5% of the total peptide pool modified, these results were eliminated from further analysis. The general heatmap highlights a significant alteration in histone modification during mineralisation (Figure 6A) and although acetylation marks were most commonly affected, methylation marks were also altered (Figure 6B,C). The frequently studied modifications of the trimethylation of H3K9 and H3K27 contribute to the stemness of mesenchymal cell populations and gene expression states [24], and it can be seen from Figure 6B that the induction of mineralisation reduced the proportion of trimethylation on H3K9, which was further reduced with SAHA. Although mineralisation reduced methylation in H3K27, the application of SAHA increased the expression of trimethylated H3K27, a finding that has been reported elsewhere [25] and attributed to a precisely timed increase in H3K27me3 at transcription start sites (Figure 6B). H1.4K25me1 and -me3 both demonstrated a classic reduction in methylation during osteogenic culture and a further decrease with SAHA (Figure 5B).

In terms of acetylation, a total of 26 acetylation modifications were analysed, of which 13 modifications were eliminated from further analysis as acetylation was less than 0.5% of the total peptide pool. Of the remaining 13 histone residue modifications, observable alterations in acetylation were in 9 out of 13 marks investigated (Figure 6D–F). The positive control HDACi (SAHA), which is expected to prevent deacetylation, increased histone acetylation in 12 out of 13 residues compared with both non-induced and induced/mineralising cultures and to a greater extent than mineralising cultures. Specific residues with changes in acetylation were on H3 (H3: K9AC, K23AC, H3.3: K36AC, and H3:K16AC) and H4 (H4: K5AC and H4: K16) (Figure 6E,F). The acetylation of K9, K14, and K27 on H3 as well as lysine 5, 12, and 16 on H4 are linked to the architectural control of chromatin [26].

## 3. Discussion

The relative expression of HDAC isoforms has been a focus of research activity in a range of tissues in the body, as Class II HDACs show controlled, tissue-specific expression while Class I HDACs are ubiquitously expressed in the nucleus of every cell in the body [13,14,15]. This has attracted attention as HDACs have a role in multiple cellular processes including mineralisation and differentiation [27], which makes them promising candidates for therapeutic intervention [1]. Within the dentine–pulp complex, the pharmacological inhibition of HDAC activity has been shown to be associated with an increase in DPC mineralisation and the expression of mineralisation-associated genes both in vitro [5,7,28] and in vivo [29]. These studies have invariably used pan-inhibitors such as trichostatin A, valproic acid, and SAHA, which indiscriminately block all Class I and II HDACs. Notably, research investigating HDACi has progressed towards the use of isoform-specific inhibitors [22], which may selectively affect the HDAC isoforms that are central to the mineralisation process rather than all isoforms. The isoform-specific inhibition of HDAC-4/5 using LMK-235 [30] and HDAC-2/3 using MI192 [31] have recently been shown to increase mineralisation in DPC cultures. Examples of isoform-specific HDACi include the aforementioned LMK235, MI192, and Tubastatin A selective against HDAC-6. However, if a strategic approach to epigenetic therapeutics is to be employed in VPT, it is important to establish the expression pattern of Class I and II HDAC isoforms in the pulp during active dentinogenesis and when crown dentine formation is complete. Therefore, one of the aims of this study was to analyse the expression of HDAC-1 to -6 in vivo and in vitro during mineralisation.

The current study highlighted that Class II HDACs are highly expressed in vivo in developing teeth, but were subsequently altered in expression in adult teeth (Figure 5), with HDAC-4 and -5 decreasing via immunohistochemical (IHC) expression and HDAC-4 gene expression significantly increased at day 14 (Figure 2D) and protein expression at day 7 (Figure 3C). By contrast, HDAC-6 showed significantly reduced expression with qPCR in mineralising DPCs and cultures (Figure 2F) with IHC, highlighting an increase in the intensity of stain in adult teeth compared with developing teeth (Figure 5). Class I HDAC expression was generally low in both developing and mature rodent molar teeth with HDAC-1 and HDAC-2 showing an increase in protein expression in mineralising DPCs, a finding only evidently corroborated in vivo for HDAC-1. Comparing these results with studies of other mineralising tissues shows similarities but also tissue-dependent differences in HDAC isoform expression. Altered HDAC expression occurs during osteogenesis [15,17], as well as dentinogenesis in third molar teeth [18], and in the periodontal ligament during cementogenesis [19]. The histological analysis of adult human molar teeth demonstrated that HDAC-2 was expressed in the dental pulp and odontoblasts, while in comparison, HDAC-1, -3, and -4 weakly were expressed within the pulp tissue [18]. Transgenic mouse knockout studies have also highlighted that HDAC-4 ablation reduced the volume of cortical bone mass [16] and altered root development [32]. Previous pharmacological HDACi studies using DPSC cultures highlighted that HDAC-3 was downregulated during odontoblast differentiation [29] and that HDAC-2 absence induced DPSC mineralisation and odontogenic gene expression [28].

The epigenetic modulation of DNA-associated proteins, including HDAC and HAT activity, alters chromatin architecture and has been shown to affect the stemness of mesenchymal SC populations [33,34,35]. A previous analysis of the nature of epigenetic modifications in DPSC and dental follicle progenitor cells (DFPCs) in mineralising cultures (DFPCs) demonstrated increased transcript levels of dentinogenic genes, including dentin matrix phosphoprotein-1 (DMP-1), and significantly induced mineralisation in DPSCs but not DFPCs [20]. The authors suggested that the two SC populations were potentially characterised during mineralisation through histone alteration. In our current study, the relative ratio of a large panel of post-translational histone modifications was analysed in mineralising culture compared with non-mineralising culture as well as a positive control of mineralised culture supplemented with the pan-HDACi SAHA. This highlighted significant alterations at day 5 between the epigenetic marks in all groups.

Although this study analysed over 80 post-translational histone modifications, certain marks have previously attracted specific attention for their effect on the differentiation of stem cell populations. For example, trimethylated H3K4 and H3K9 are considered marks of a heterochromatin state, the condensed repressive chromatin structure, and it has been shown that the repressive nature of trimethylated H3K4 and H3K9 is lost during adipogenic differentiation [36]. Furthermore, in DPC cultures when trimethylated H3K27 marks were absent on DMP-1 and DSPP promoters, gene expression was considerably higher [20]. As a result, trimethylated HSK27, in addition to H3K4 and H3K9, are the most investigated histone modifications for their role in the differentiation of stem cells as well as their effects on transcription [24], in which dental periodontal cell populations highlight a switch from trimethylated H3K4 to H3K9 during osteogenic differentiation [37]. Within the current study, there are clear alterations in trimethylation marks during DPC mineralisation with decreases in the proportion of H3K9me3, H3K27me3, and H12K5me3 after osteogenic induction (Figure 5B,C) that confirm previous findings in dental and non-dental cell populations [36,37].

Specifically, in terms of acetylation, H3 and H4 are the most studied modifications with the acetylation of lysine 9, 14, and 27 on H3 as well as lysine 5, 8, 12, and 16 on H4 associated with the control of the chromatin architecture [26,38] (Figure 6D–F—green box). Certain acetylation marks, for example, H4K16ac, open up the chromatin structure, while other marks such as H4K20ac have not been shown to alter chromatin compaction [39]; this is also evident in our current study where H4K20ac was not altered (Figure 6A). Notably, in contrast to trimethylation modifications, acetylation marks are associated with increased gene expression and mineralisation effects as evidenced by the increase in mineralisation after the application of HDACi in DPC populations [7,21,29]. In our present study, the application of a pan-HDACi (SAHA) as a positive control, as expected, increased the peptide pool of acetylation in the vast majority of marks. These findings highlight the importance of histone modification and epigenetic control on the differentiation of DPCs during mineralisation.

Current VPT materials, such as hydraulic calcium-silicate cements, do not predictably regenerate dentine [40] and are not designed to target a specific biological process [1,2]. This leads to a lack of control over the nature and quality of the repair processes. For this reason, the role of epigenetic regulation and the potential therapeutic targeting of these processes, clinically using pharmacological inhibitors, are particularly attractive. This study for the first time has highlighted that as DPCs mineralise, there are multiple epigenetic changes evident in these cells, inducing alterations that affect the architecture of chromatin, inducing gene expression and pulpal repair processes [7]. Furthermore, the expression of selected HDAC isoforms significantly modifies during active phases of mineralisation, paving the way for the use of isoform-selective HDACi potentially focused on HDAC-4 or -6 to promote dentinogenesis and odontoblast activity. The particular advantage of this from a clinical VPT perspective is that this targeted therapy can be carried out topically and incorporated into a next-generation dental material to alter odontoblast activity in a controlled manner.

This study does have some limitations, including the use of rat incisor pulp tissue for cell culture studies. Although the use of rodent incisors has been employed by our group and others in previous publications [3,7], the rodent incisor, unlike a human tooth, exhibits a pattern of continuous eruption throughout life, which may affect characteristics such as cell proliferation or signalling in downstream assays. Rat pulp is clearly not human, and differences in signalling or response to mineralisation cues may differ; however, within this study, as closely standardised tissues were required to monitor gene and protein expression in vitro, the use of a rodent model was preferred throughout.

In conclusion, our results highlight for the first time the expression of Class I and II HDAC isoforms during both in vitro mineralisation and in vivo dentinogenesis. Class II HDACs, including HDAC-4, -5, and -6, show distinct differences in expression, which could make them attractive targets in the future for therapeutic intervention. Finally, it is evident that there is a dynamic alteration in histone acetylation occurring during mineralisation, which is amplified with SAHA application with a decrease in the transcriptionally repressive trimethylation marks and an increase in the relaxed chromatin-associated acetylation marks, which reflects the importance of epigenetic control on DPC mineralisation and differentiation.

## 4. Materials and Methods

### 4.1. Animals

All approved tissue use from Wistar Hannover rats was in accordance with the relevant guidelines of the Ethical Guidelines Committee of the Comparative Medicine Unit, Trinity College Dublin under individual authorisation (AE19136/I049). All animal work and husbandry complied with the Animal Research: Reporting in Vivo Experiments (ARRIVE) guidelines.

#### In Vivo Model and Immunohistochemical (IHC) Staining

The maxillary bones were dissected from postnatal (5 days old) and adult (8 weeks old) Wistar Hannover rats. The specimens were fixed in 10% buffered formalin (Sigma-Aldrich, Arklow, Ireland) for 24 h and demineralised in 10% EDTA (Sigma-Aldrich, Arklow, Ireland). The fixed tissue was dehydrated through ascending concentrations of ethanol, embedded in paraffin (Leica, Sheffield, UK), and serial sections of 5 µm thickness were prepared. To detect the expression and distribution of Class I and II HDAC markers, IHC was carried out on similar sections using the Envision+ Horseradish Peroxidase (HRP) staining system (Dako-Agilent, Cork, Ireland) according to the manufacturer’s instructions. Briefly, the sections were treated with an enzyme blocker for 10 min (Dako) and nonspecific binding to proteins was blocked using 1% bovine serum albumin (BSA) (Sigma-Aldrich, Arklow, Ireland) + 2% goat serum (SantaCruz Biotechnologies, Dallas, TX, USA) for 1 h before incubation with primary antibodies. The following antibodies were used: anti-HDAC-1, anti-HDAC-2, anti-HDAC-3, anti-HDAC-4, anti-HDAC-5, anti-HDAC-6 (all 1:100, Abcam, Cambridge, UK), or normal immunoglobulin G (IgG) as negative control; this was performed overnight at 4 °C. The samples were incubated with labelled HRP-polymer for 30 min at room temperature. For final visualisations, the samples were developed with chromogen (Dako-Agilent, Cork, Ireland) and counterstained with haematoxylin (Sigma-Aldrich, Arklow, Ireland). Indicated specimens were incubated in sodium citrate buffer (10 mM citric acid, pH 6.0) for 1 h at 60 °C for antigen retrieval and before proceeding to the staining. Serial sections were examined from 5 rats at both postnatal (*n* = 5) and adult (*n* = 5) timepoints (Table 1).

### 4.2. Dental Pulp Cell Studies

#### 4.2.1. Primary Cell Isolation and Culture

The primary DPCs used in this study were isolated from the pulpal tissue of freshly extracted rodent incisor teeth using an enzymatic disaggregation technique [7] (Table 1). All teeth were dissected from male Wistar Hannover rats aged 25–30 days and weighing between 120 and 140 g. Post-extraction, the pulp tissue was extirpated from the teeth and minced aseptically into pieces of approximately 1.0 mm^2^. The tissue was transferred into a 50 mL Falcon™ tube (BD Biosciences, Oxford, UK) containing 4 mL of Hank’s balanced salt solution (Sigma-Aldrich, Arklow, Ireland) (2.5 g/L of trypsin, 0.2 g/L of ethylenediaminetetraacetic acid [EDTA], and 4Na) prior to incubation at 37 °C at 95% relative humidity in a 5% CO_2_ atmosphere for 45 min (MCO-18AC incubator, Sanyo Electric, Osaka, Japan). In order to improve dissociation, the DPCs were constantly agitated during digestion (MACSmix tube rotator, MiltenyiBiotec, Surrey, UK) prior to an equal volume of supplemented α-MEM (Biosera, East Sussex, UK) containing 1% penicillin/streptomycin (Sigma-Aldrich, Arklow, Ireland) and 10% (*v*/*v*) foetal calf serum (FCS) (Biosera, East Sussex, UK) being added. In order to obtain a single cell suspension, the digest was passed through a 70 µm cell sieve (BD Biosciences, Wokingham, UK) and the cells were pelleted via centrifugation at 1200 rpm for 3 min, prior to re-suspension in 1 mL of supplemented α-MEM. Subsequently, the DPCs were expanded in culture to passage 3 for use in all subsequent experiments with the media changed every 3 days. Unless otherwise stated, all cultures were maintained in supplemented α-MEM at 95% relative humidity in a 5% CO_2_ atmosphere at 37 °C.

#### 4.2.2. Histone Deacetylase Inhibitor Preparation

A 5 mM stock solution of the HDACi, suberoylanilide hydroxamic acid (SAHA [N-hydroxy-N0-phenyl-octanediamide]) (Sigma-Aldrich, Arklow, Ireland), in dimethyl sulfoxide (DMSO), was diluted in phosphate-buffered saline (PBS) (Sigma-Aldrich, Arklow, Ireland) prior to further dilution to an experimental concentration with supplemented a-MEM.

#### 4.2.3. Mineralisation Assay

DPCs were seeded (5 × 10^4^ per well in 6-well plates) and cultured in supplemented α-MEM for 72 h until 60–70% confluent. At 72 h (day 0), the DPCs were cultured in a mineralisation medium containing supplemented α-MEM including 50 µg/mL of ascorbic acid, 0.1 mM of dexamethasone, and 10 mM of β-glycerophosphate. The HDACi-exposed group was cultured in a mineralisation medium with 1 µM of SAHA (Sigma-Aldrich) only for the first 3 days [16] prior to culture with an HDACi-free mineralisation medium for the duration of the experimental period up to 14 or 21 days. These timepoints were selected as rat DPCs in mineralising culture require at least 14 days to secrete quantitative minerals that can be discriminatively measured using alizarin red staining [7]. The control group was in supplemented α-MEM or in a mineralisation medium without HDACi throughout the period. For all the groups, the medium was changed every 3 days. An equivalent volume of DMSO/PBS was added to the wells in which there was no HDACi. During the experiment period, alizarin red staining was carried out at each timepoint to survey the mineralisation. For alizarin red analysis, DPC cultures were washed three times in PBS, fixed in 10% formaldehyde (Sigma-Aldrich, Arklow, Ireland) for 15 min, washed with distilled water, and stained with 1.37% (*w*/*v*) alizarin red solution (pH 4.2) (Millipore, Cork, Ireland) for 15 min at room temperature. Excess stain was removed by washing with distilled water. Three independent experiments (*n* = 3) were performed in triplicate for all mineralisation assays.

#### 4.2.4. Western Blotting

Analysis was carried out on cell lysates from DPCs cultured in mineralisation media for 0, 1, 4, 7, 11, 14, and 21 days. Protein lysates were prepared by incubating the cells for 30 min at 4 °C in extraction buffer, comprising RIPA buffer, phenylmethyl sulfonyl fluoride (PMSF), and halt protease inhibitor (all from ThermoFisher Scientific, Waltham, MA, USA) prior to centrifugation at 13,000 rpm for 10 min at 4 °C. Total protein was quantified via a Bradford assay (Bio-Rad Laboratories GmbH, München, Germany) using the NanoDrop 2000 spectrophotometer (ThermoFisher Scientific, Dublin, Ireland) before 30 µg proteins were used for sodium dodecyl sulphate–polyacrylamide gel electrophoresis (SDS-PAGE) on a 4–20% polyacrylamide gel (Mini-PROTEAN, Bio-Rad, Hercules, CA, USA) and the migrated proteins were transferred to a polyvinylidene difluoride (PVDF) membrane. To block nonspecific protein binding, the membrane was incubated in 5% non-fat dry milk at room temperature for 1 h prior to overnight incubation at 4 °C with primary antibodies raised against HDAC-1, -2, -3, -4 (#9928), -5 (#20458), and -6 (#7612, All Cell Signaling Technology, Leiden, The Netherlands) and β-actin (#3700, Cell Signaling Technology, Danvers, MA, USA) in a 1:1000 solution in 2.5% skimmed milk. After washing, the membrane was incubated with 2ry antibodies at room temperature for 2 h, developed with an enhanced chemiluminescence (ECL) detection kit (Clarity Western, Bio-Rad, Hercules, CA, USA), and then detected with a Chemidoc Touch Imaging System (Bio-Rad, Hercules, CA, USA). The protein expression was evaluated via densitometric analysis (Bio-Rad, Hercules, CA, USA) with β-actin used as a loading control to normalise the data. Seven independent experiments (*n* = 7) were carried out for each HDAC target.

#### 4.2.5. RNA Isolation and cDNA Synthesis

DPCs were seeded at 5 × 10^4^ per well in six-well plates and cultured in supplemented α-MEM for 72 h until 60–70% confluent. At 72 h (experimental day 0), the cells were cultured in a supplemented mineralisation medium only for the duration of the experimental period or in cultures containing 1 μM of SAHA for 72 h before incubation with an HDACi-free mineralising medium until cell harvest. Control samples contained cells cultured in a mineralisation medium without HDACi. Cultures were detached with trypsin/EDTA (Sigma-Aldrich, Arklow, Ireland), homogenised for 30 s using a T10 basic S2-Ultra-Turrax tissue disrupter (IKA, Staufen, Germany), and had their RNA extracted using the RNeasy mini kit (Qiagen, West Sussex, UK) according to the manufacturer’s instructions. RNA was isolated on the RNAeasy mini-column assembly (Qiagen, West Sussex, UK) prior to conversion to single-stranded cDNA using the TaqMan reverse transcriptase kit (Applied Biosystems, Waltham, MA, USA). cDNA concentrations were determined spectrophotometrically at 260 nm (Nanodrop 2000, ThermoFisher Scientific, Dublin, Ireland).

#### 4.2.6. Reverse Transcriptase Polymerase Chain Reaction (qRT-PCR)

The qRT-PCR analysis was performed for rat genes using specific primers (Invitrogen, Life Technologies, Thermo Scientific, Dublin, Ireland). Primer sequences, accession numbers, and product sizes are listed (Appendix A). A total of 100 ng of synthesised cDNA was amplified in a reaction containing 12.5 µL of Fast SYBR Green Master Mix (Applied Biosystems, Waltham, MA, USA), 8 µL of DPEC-treated water, and Forward/Reverse primers, and then was detected using the Applied Biosystems 7500 Fast Real-Time PCR thermal cycler (Applied Biosystems, Waltham, MA, USA). The thermal cycler was subjected to a designated number of amplification cycles (40 cycles), where a typical cycle was 95 °C for 3 s and 60 °C for 30 s. Real-time PCR data were normalised to β-actin and fold change in gene expression was obtained using the formula 2^((Ctctrl − Ctβ-actin) − (Ctexp − Ctβ-actin))^, where *Ct* is the threshold cycle, *ctrl* is the control, and *exp* is the experimental samples. Four independent experiments (*n* = 4) were carried out in triplicate for the target genes. The MIQE guidelines were used for the qRT-PCR data reporting.

#### 4.2.7. Mass Spectrometric Analysis of Post-Translational Modifications in Mineralising DPCs

DPCs (from passage 3) were cultured in either supplemented α-MEM (control), mineralisation media, or mineralisation media supplemented with 1 µM of SAHA for 5 days in T25 cell culture flasks (Sarstedt, Wexford, Ireland). The control group was either supplemented media or mineralisation media without SAHA; 2–3 × 10^6^ DPCs were detached with trypsin-EDTA (Sigma-Aldrich, Arklow, Ireland) and centrifuged at 800 rpm for 5 min. The cell pellets were washed in PBS and over 80 post-translational histone modifications (including 13 related to acetylation) were analysed using mass spectrometry (Mod Spec, Active Motif, Carlsbad, CA, USA) (Appendix A). Briefly, histones were extracted via acid, derivatised using propionic anhydride (Sigma-Aldrich, Arklow, Ireland), and digested with trypsin as previously described [41] before being measured on three separate occasions using the TSQ Quantum Ultra triple quadrupole mass spectrometer (ThermoFisher Scientific, Dublin, Ireland) coupled with an UltiMate 3000 Dionex nano-liquid chromatography system (ThermoFisher Scientific, Dublin, Ireland). The data were quantified using Skyline as previously described [42] and represent the percentage of each modification within the total pool of that amino acid residue.

### 4.3. Statistical Analysis

One-way analysis of variance (ANOVA) and Tukey’s post hoc tests were used to determine the expression of HDAC compared with control at various timepoints using Western blotting and qRT-PCR (*p* < 0.05) (SigmaStat 20.0 software (SPSS, IBM, IL, USA)). The statistical significance of qRT-PCR data was also assessed using Student’s *t*-test.

## Figures and Tables

**Figure 1 ijms-25-06569-f001:**
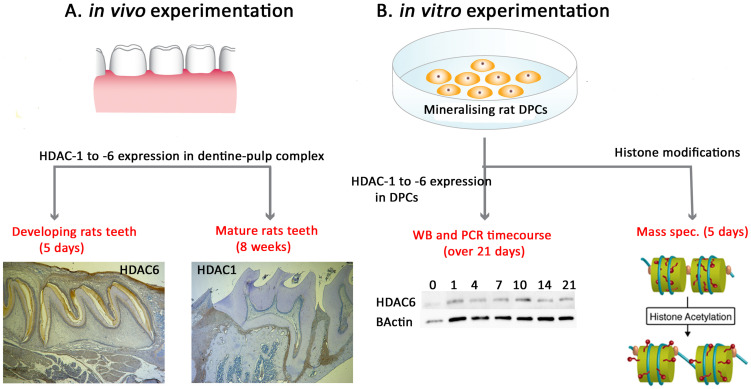
Schematic of experimental design and project flow.

**Figure 2 ijms-25-06569-f002:**
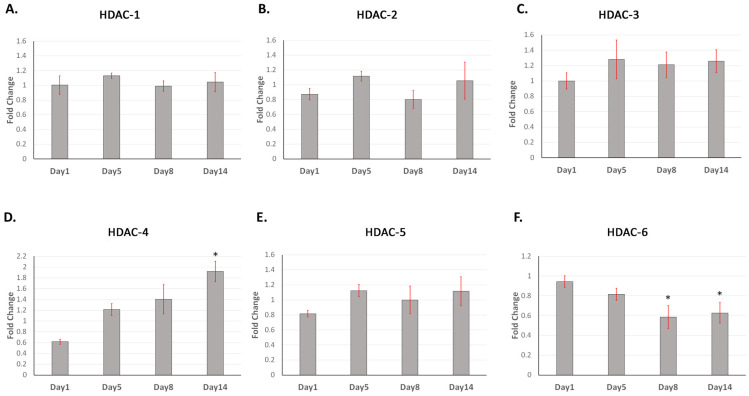
Gene expression time-course analysis confirms alterations in the expression of Class II HDAC isoforms during DPC mineralisation. Relative gene expression, using quantitative real-time PCR, of DPCs cultured in a mineralisation medium was analysed relative to day 0 expression and presented as fold change. Day 0 corresponds to a 1-fold change on each panel graph. Significant differences related to *HDAC-4* which was significantly increased (**D**) while *HDAC-6* was decreased at day 14 (**F**). (**A**) *HDAC-1*; (**B**) *HDAC-2*; (**C**) *HDAC-3*; (**D**) *HDAC-4*; (**E**) *HDAC-5*; and (**F**) *HDAC-6*. Β-actin was used as an internal control. Error bars represent means ± SEM of three independent biological samples carried out in triplicate. One-way analysis of variance (ANOVA) and post hoc Tukey’s test were used to ascertain statistical significance at *p* < 0.05 *.

**Figure 3 ijms-25-06569-f003:**
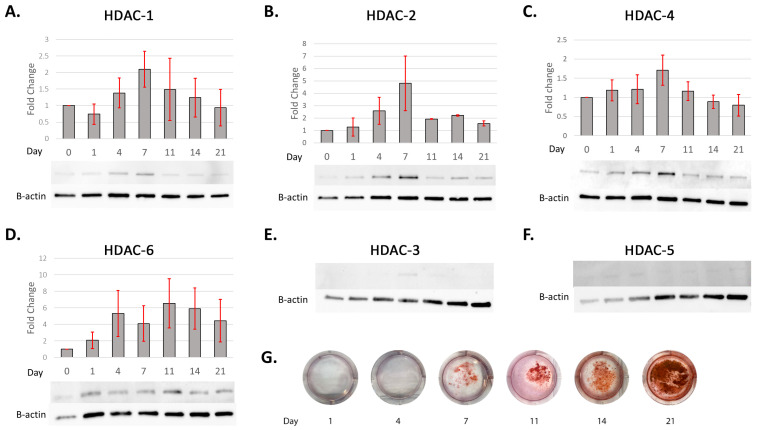
Time-course analysis of HDAC expression during DPC mineralisation showing altered expression of HDAC isoforms. HDAC expression of DPCs cultured in a mineralisation medium. HDAC-1, -2, and -4 showed increased expression until day 7 and then decreased (**A**–**C**) while HDAC-6 remained relatively high after day 4 (**D**). Images are representative of Western blotting result from each HDAC isoform (upper) and its internal control (β-actin; lower). Expression levels were normalised and presented as fold changes against day 0 expression. (**A**) HDAC-1; (**B**) HDAC-2; (**C**) HDAC-4; (**D**) HDAC-6. The expression levels of HDAC-3 and -5 were too weak to evaluate with all the samples tested (**E**,**F**). Error bars represent means ± SEM of at least three independent biological replicates. Alizarin red staining of DPC mineralisation-induced culture shows the mineralisation starts around day 7 and continues until day 21 (**G**).

**Figure 4 ijms-25-06569-f004:**
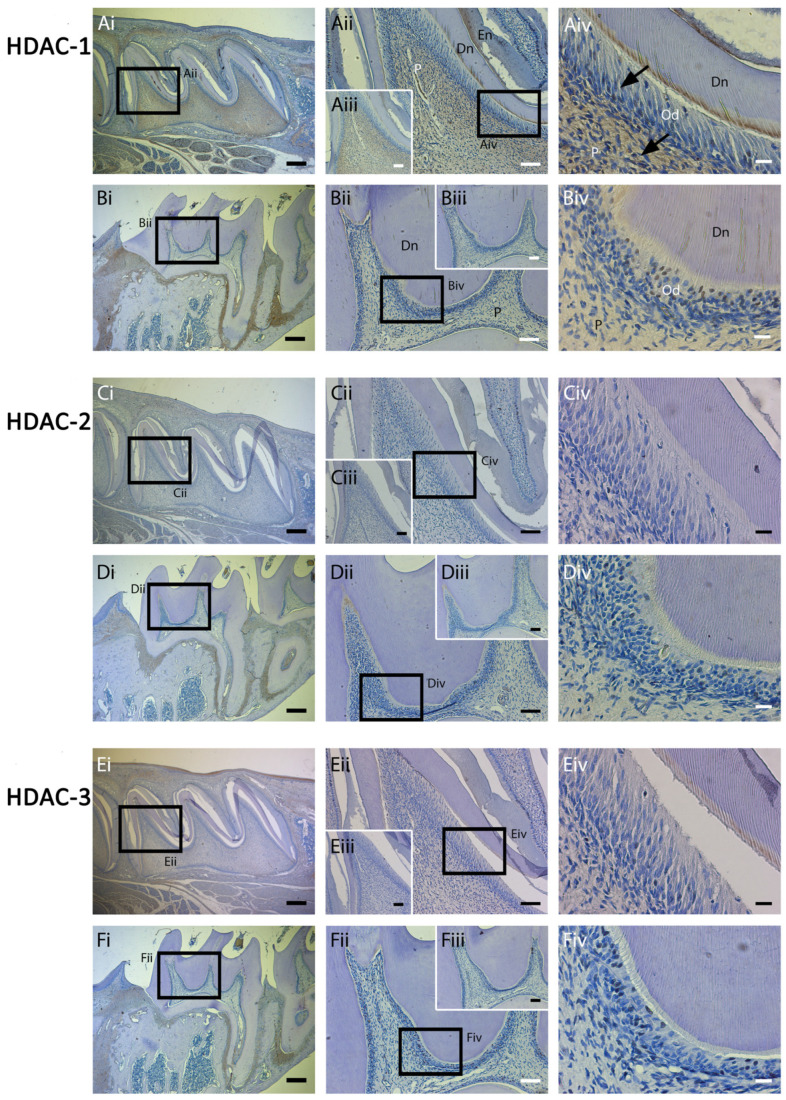
Class I HDAC isoforms exhibit altered expression in postnatal and adult molar teeth. Comparison of Class I HDAC expression in rat molars at postnatal 5 days (first maxillary molar, (**A**,**C**,**E**)) and 8 weeks (third maxillary molar, (**B**,**D**,**F**)). (**A**,**B**) HDAC-1 is highly expressed in postnatal dental pulp, odontoblasts ((**A**)—black arrows), and ameloblasts compared to negative control (Aiii) but reduced in adult teeth (**B**). (**C**,**D**) Expression of HDAC-2 was low, not being evident in both developing (**C**) and mature (**D**) teeth. (**E**,**F**) Meanwhile, HDAC-3 expression was also low in both developing (**E**) and mature (**F**) teeth. HDAC-1 to 3 antibody-positive cells demarcated in brown in individual panels. (**Aii**–**Fii**) represent higher magnifications of the black boxes in (**Ai**–**Fi**), respectively. (**Aiv**–**Fiv**) represent higher magnifications of the black boxes in (**Aii**–**Fii**), respectively. Inset images (**Aiii**–**Fiii**) represent negative controls of each respective experiment. Scale bars = (**Ai**–**Fi**) 200 μm (original mag. ×2.5), (**Aii**–**Fii**,**Aiii**–**Fiii**) 50 µm (original mag. ×10), and (**Aiv**–**Fiv**) 10 µm (original mag. ×40). Abbreviations: P, pulp tissue; Dn, dentine; Od, odontoblasts; En, enamel. Multiple sections were examined from 5 independent rats at 5 days and 8 weeks, respectively.

**Figure 5 ijms-25-06569-f005:**
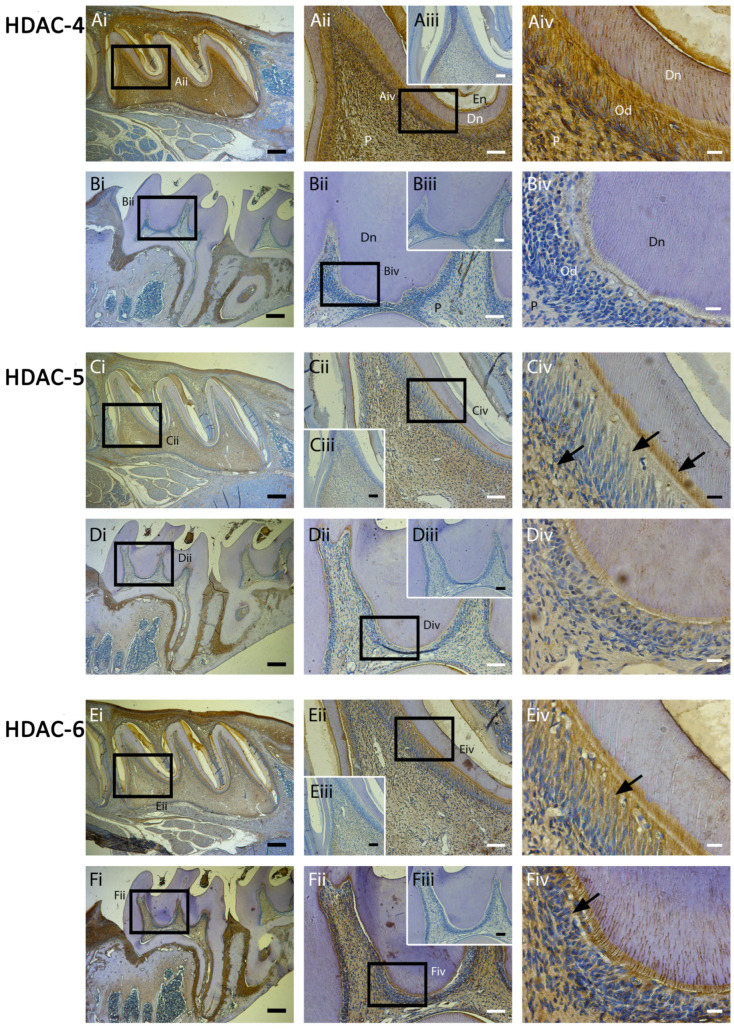
Class II mineralisation-associated HDAC isoforms exhibit high odontoblast and dental pulp cell (DPC) expression, which alters between developing and adult molars. Comparison of Class II HDAC expression in rat molars at postnatal 5 days (first maxillary molar,(**A**,**C**,**E**)) and 8 weeks (third maxillary molar, (**B**,**D**,**F**)). (**A**,**B**) HDAC-4 was strongly expressed in odontoblasts, ameloblasts, and DPCs in developing teeth (**A**) but weakly expressed in mature teeth. (**C**,**D**) HDAC-5 was relatively highly expressed in the dental pulp, odontoblast, and predentine of developing teeth ((**C**)—black arrows) as well as mature teeth (**D**). (**E**,**F**) HDAC-6 was highly expressed within the dentine–pulp complex of both developing (**E**) and further increased in mature (**F**) teeth, with particularly high expression in the odontoblast layer ((**E**,**F**)—black arrows). HDAC-4 to -6 antibody-positive cells demarcated in brown in individual panels. (**Aii**–**Fii**) represents higher magnifications of the black boxes in (**Ai**–**Fi**), respectively. (**Aiv**–**Fiv**) represents higher magnifications of the black boxes in (**Aii**–**Fii**), respectively. Inset images (**Aiii**–**Fiii**) represent the negative control of each experiment. Scale bars = (**Ai**–**Fi**) 200 μm (original mag. ×2.5), (**Aii**–**Fii**,**Aiii**–**Fiii**) 50 µm (original mag. ×10), and (**Aiv**–**Fiv**) 10 µm (original mag. ×40). Abbreviations: P, pulp tissue; Dn, dentine; Od, odontoblasts; En, enamel. Sections were examined from 5 rats at 5 days and 8 weeks, respectively.

**Figure 6 ijms-25-06569-f006:**
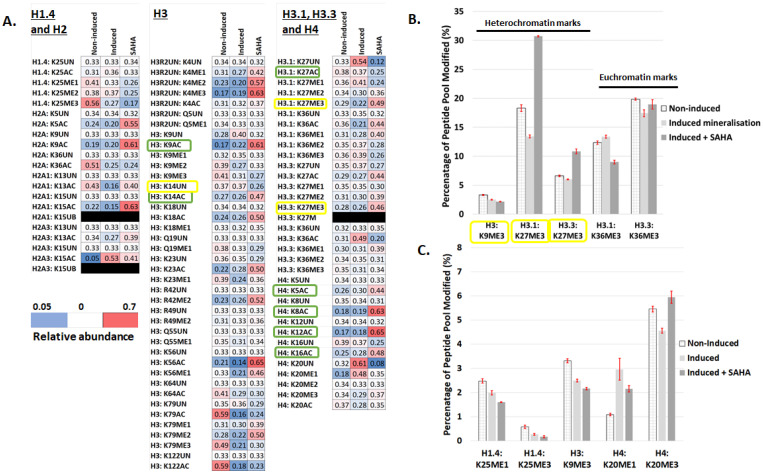
Mineralisation of DPC cultures dynamically alters multiple histone modifications (**A**) Heatmap representing 80 individual histone modifications (including unmodified peptides) analysed on H1.4, H2, H3, H3.1, H3.3, and H4. The data for each sample are converted to the fraction of the sum across all samples to display relative abundances across the 3 sample groups (non-mineralising, mineralising, and mineralising + SAHA). Red: high abundance compared with other samples and blue: low compared to other samples. This approach ignores the absolute levels of the modifications. Heterochromatin-associated marks are circled in yellow and highlighted in (**B**). (**C**) Other altered methylation marks of interest are also highlighted. (**A**,**D**–**F**) Thirteen acetylation marks illustrated for 3 groups with the acetylation of lysine associated with the control of the chromatin architecture are highlighted with a green box.

**Table 1 ijms-25-06569-t001:** Inclusion criteria for teeth used in the study. DPC = dental pulp cell.

Technique	Experimental Teeth Used		
	Developmental In Vivo	Adult In Vivo	Mineralisation Assay In Vitro
Histology	5-day-old Wistar Hannover rats, maxillary 1st molar teeth	8-week-old Wistar Hannover rats, maxillary 1st and 3rd molar teeth	
DPC Culture			Male Wistar Hannover rats aged 25–30 days and weighing between 120 and 140 g, maxillary and mandibular incisor teeth

## Data Availability

Data is contained within the article and Appendix A.

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
