# Peer review of "Dynamic Alterations in Acetylation and Modulation of Histone Deacetylase Expression Evident in the Dentine–Pulp Complex during Dentinogenesis"

_ijms, 2024, doi:10.3390/ijms25126569_

Round 1
Reviewer 1 Report
Comments and Suggestions for Authors
The manuscript entitled “Dynamic alterations in acetylation and modulation of Histone Deacetylase expression evident in the dentine-pulp complex during dentinogenesis” is a comprehensive pre-clinical study involving gene expression, protein detection and epigenetics regarding dental pulp mineralization. The reviewer believes that structural improvements and an adequate clinical contextualization are needed. Below, authors can find comments and suggestions.
Minor to moderate language revision is needed. Additionally, there are US and UK English language terms, requiring an uniformization (e.g., mineralisation/mineralization; analyse/analyze).
Add the meaning of IHC abbreviation.
Abstract:
Adequate, succinct, and well-structured.
Introduction:
There is some room for improvement, particularly, regarding the clinical application of the generated knowledge. In the reviewer’s opinion, there are too many details about the molecular theory itself, as definitions and concepts. Authors could summarize this part (e.g., Line 53 – 72) and explore why choosing this therapy over the established ones, or how such knowledge could be applied to the current ones. Moreover, pulp mineralization can be a welcomed effect in clinical applications, but also can be pathological and deleterious. The introduction did not make clear which path the authors are focused on, and the scenarios that an induced mineralization would be welcomed, as pulp capping, incomplete rhizogenesis or healing of periapical lesions, or scenarios that a premature pulp mineralization could be harmful, and, therefore, be avoided.
Line 48 – 53: In the reviewer’s opinion, the information of this paragraph could be removed.
Methods:
Authors used dental pulp cells from rat incisors in some of the assays. These teeth present important differences from human incisors, namely, continuously growth over time and natural protrusion. Rat molars, on the other hand, can be considered more similar, in line with the IHC used in the present study. Since these differences can impact in the cellular signaling, renewing and overall function, justify the choice of cells from incisor teeth, or address this limitation (in the discussion section).
Authors should follow the ARRIVE guidelines to describe the animal care prior dental extractions.
Cellular experiments with SAHA: Does the control group receive the same amount of DMSO of experimental groups?
4.15/4.16: Authors should follow the MIQE guidelines for describing PCR assessment. Relevant information is missing.
4.2. Add the number of the approval of the animal study protocol. Moreover, since the cellular assays were set on primary cell cultures extracted from the animal, 4.2. should be placed before the cellular description.
Results:
Since this section is placed before the methodology, the reviewer struggled to understand the experimental models that were assessed and described. See the comment below to address it.
2.1. section is confusing, authors mixed in vivo and in vivo assessments, protein, and gene expression. As a suggestion, authors could add a paragraph before 2.1. describing what was performed, and the order that the results are organized. Moreover, since the experimental model is slightly different (in vitro and in vivo), they should be totally separate.
Figure 1, A to D, disclaim (in the legend) if a statistical test was performed and non-significant differences were observed between timepoints. Authors performed a semi-quantitative analysis of WB. Why not applying the same methodology for the AR staining, using ImageJ or similar software?
Figure 2: Authors describe the gene expression between timepoints of cells cultured in osteogenic/mineralization medium, and graphs were normalized by day 0. However, the reviewer missed the basal culture data, which could add relevant information regarding the mineralization induction over the expression of these genes. By reading the section 4.1.5., the reviewer expected to find the PCR analysis of cells cultured in SAHA in addition to the mineralization medium and a gene expression comparison between both conditions. Please, explain.
Since gene expression precedes protein synthesis, would be more consistent to display them first. Add to the legend that day 0 corresponds to 1-fold change in the graphs.
2.2. Authors competently described in the text figure 3. As a suggestion, add the description of the colors of the marked protein/anti-body.
2.3. Figure 4 is displaced. As in the previous comment, the expected color of the protein/anti-body should be added in the legend.
2.4. Explained the motive for the timepoint selection, good job.
Discussion:
Authors discussed the molecular alterations and tried to associate the results from gene expression, protein detection, and the histone modifications observed. However, the clinical aspect of the results in lacking, in the reviewer’s opinion, that is, how this knowledge could be applied in the clinical scenario, and the advantages of using epigenetics over traditional methods, as the reviewer commented on the introduction section.
Comments on the Quality of English LanguageMinor to moderate language revision is needed. Additionally, there are US and UK English language terms, requiring an uniformization (e.g., mineralisation/mineralization; analyse/analyze).
Author Response
Rebuttal to line one attached.

Reviewer 2 Report
Comments and Suggestions for Authors
Dear authors, thank you for submitting the manuscript "Dynamic alterations in acetylation and modulation of Histone-Deacetylase expression evident in the dentine-pulp complex during dentinogenesis". I enjoyed reading your study and please know that my comments are intended to be constructive.
-Please provide a graphic abstract (workflow chart) so readers can easily follow the sequence of your study and your results. You should insert this chart in the results section.
-For materials and methods, please include a table with the inclusion criteria of the specimens (teeth) selected for the study.
-For the introduction, provide hypotheses or qualitative statements about the effects of the expected results of the study.
-Expand the limitations of the study in the discussion section.
-iThenticate report displays 34%, please decrease it, the following articles have high number of words similar to your manuscript: https://doi.org/10.3389/fcell.2022.883266, https://doi.org/10.3390/
ijms241210403 and https://doi.org/10.3390/
ijms24054814.
-In some sections such as 4.1.3 and 4.1.4 consider including tables in order to summarize the materials/techniques used.
Author Response
Line by line response to reviewer 2 attached.

Round 2
Reviewer 1 Report
Comments and Suggestions for Authors
The authors politely addressed all questions and concerns of this reviewer. Congratulations.
Comments on the Quality of English LanguageMinor editing of English language required